# Inequities in Screening and HPV Vaccination Programs and Their Impact on Cervical Cancer Statistics in Romania

**DOI:** 10.3390/diagnostics13172776

**Published:** 2023-08-28

**Authors:** Laurentiu Simion, Vlad Rotaru, Ciprian Cirimbei, Laurentia Gales, Daniela-Cristina Stefan, Sinziana-Octavia Ionescu, Dan Luca, Horia Doran, Elena Chitoran

**Affiliations:** 1“Carol Davila” University of Medicine and Pharmacy, 050474 Bucharest, Romania; laurentiu.simion@umfcd.ro (L.S.); laurentia.gales@umfcd.ro (L.G.); cristinastefan10@gmail.com (D.-C.S.); luca_dan94@yahoo.com (D.L.); horia.doran@umfcd.ro (H.D.); chitoran.elena@gmail.com (E.C.); 2General Surgery and Surgical Oncology Department I, Bucharest Institute of Oncology “Prof. Dr. Al. Trestioreanu”, 022328 Bucharest, Romania; 3Medical Oncology Department, Bucharest Institute of Oncology “Prof. Dr. Al. Trestioreanu”, 022328 Bucharest, Romania; 4Surgical Clinic I, Clinical Hospital Dr. I. Cantacuzino Bucharest, 030167 Bucharest, Romania

**Keywords:** social and global inequities, access to health programs, cervical cancer screening program, HPV vaccination programs, knowledge, believes, acceptance of vaccines/screening, participation in screening programs, incidence and mortality rates for cervical cancer, pelvectomies

## Abstract

(1) Introduction: A Romanian woman is diagnosed with cervical cancer every two hours; the country ranks second in Europe in terms of the mortality and incidence rate of this disease. This paper aims to identify the main reasons that have led to this situation, focusing on the measures taken by the Romanian Ministry of Health for the prevention of this type of cancer—national programs for cervical cancer screening and HPV vaccination. (2) Materials and methods: We performed a study based on the available secondary data from the National Statistics Institute, World Health Organization and Bucharest Institute of Oncology in order to assess the burden associated with cervical cancer and place it in the context of known global and European incidence and mortality rates, thus evaluating the importance of this health issue in Romania. The second component of our study was a cross-sectional study. Here, we used a 14-question questionnaire applied to the women participating in the National Screening Program for Cervical Cancer and aimed to evaluate the women’s level of knowledge about screening and HPV vaccination and their access cervical-cancer-specific healthcare services. (3) Results: The results of this research show that a high percentage of women postpone routine checks due to a lack of time and financial resources and indicate that a low level of knowledge about the disease and the specific preventive methods determines the low participation in screening and HPV vaccination programs implemented in Romania, contributing to the country’s cervical cancer situation. (4) Conclusions: The national programs have complicated procedures, are underfunded and do not motivate healthcare workers enough. This, combined with the lack of information for the eligible population, adds up to an extremely low number of women screened and vaccinated. Our conclusion is that the Romanian Ministry of Health must take immediate action by conducting major awareness campaigns, implementing measures to make the programs functional and ensuring coherent funding.

## 1. Introduction

A Romanian woman is diagnosed with cervical cancer every two hours [1]; the country ranks second in Europe in terms of the mortality and incidence rate of this disease [2]. In Romania, neoplastic diseases represent the second most common cause of death, after cardiovascular disease [3], and cervical cancer remains by far one of the most severe problems faced by the healthcare system. If when it comes to other cancers, Romania follows the European trend; in the case of cervical cancer, Romania ranks second in Europe (behind only Montenegro) in terms of both the incidence and specific mortality [2]. In Romania, the incidence rate of cervical cancer is more than double the average in European countries (34.2 vs. 15.0), and the mortality rate is triple (18.3 vs. 6.7). These incidence values situate the country among low-income countries or countries where long-lasting armed conflicts took place, making the healthcare issues secondary and ranking 43rd worldwide [2]. A total of 52–60% of the deaths caused by cervical cancer occur in low- and middle-income counties due to a failure in implementing population-based screening programs [4,5,6], and Romania stands out in a negative way as it is an upper middle-income country [7].

The situation in Romania is unnecessary, especially in the context of a disease with a very well documented infectious etiology, an extremely effective vaccine against the main etiological factor for the disease and for which we have countless screening methods adapted to any healthcare budget that proven effectiveness over a number of years in other countries. Currently, the cervical-cancer-specific preventive measures in Romania are comprised of a national screening program (addressed to women aged 25 to 64 with no prior cervical cancer diagnosis or other known precursor cervical lesions) and a non-gender-neutral national HPV vaccination program (addressed to girls aged 11–18). The screening program functions on a voluntary basis, is not mandatory and no invitations to attend are sent to eligible women. The HPV vaccination program is available at the family doctor. The vaccine (Gardasil-9) is not mandatory and is not part of the National Vaccination Program. The full three-dose course is available free of charge; however, an active request for the vaccine by the parents of the child in question and their consent for the vaccination is required.

This paper aims to identify the main reasons that have led to this situation, focusing on the measures taken by the Romanian Ministry of Health for the prevention of this type of cancer—national programs for cervical cancer screening and HPV vaccination—and on the characteristics of Romania’s population that may contribute to the present state.

## 2. Materials and Methods

Our study has two components. One is based on descriptive statistical information from the internal electronic database of Bucharest Institute of Oncology “Prof. Dr. Alexandru Trestioreanu”, the largest center for cancer care in Romania. In Romania, there are 3 major oncology institutes (in Bucharest, Cluj-Napoca and Iasi), each being the primary facility for cancer-related treatments for about a third of the territory of Romania and for a population of around 8 million (out of the 22 million resident citizens of Romania). We tried to assess the burden associated with cervical cancer among the pathologies treated in our hospital and placed it in the context of known global and European incidence and mortality rates, thus evaluating the importance of this health issue in Romania. Permission for retrieving data from the internal electronic database and statistical analysis was obtained from the leadership of the institute. For this study, we screened the electronic database of our institution using as filtering criteria ICM 10 code C53—Malignant tumor of cervix—and extracted data concerning the number of patients treated each year for cervical cancer between 2012 and 2022, the number of distinct hospitalization episodes that were necessary for cervical cancer, the number of surgical procedures performed for this pathology and the total number of surgical procedures performed.

The second component was a cross-sectional study, using a questionnaire as a research tool that was applied to the women participating in the National Screening Program for Cervical Cancer during Pap smear testing campaigns with the mobile screening unit that were organized in disadvantaged parts of Romania (known for lower income and accessibility to education and healthcare). All participants signed an informed consent form. The questionnaire contained 14 questions presented in Table 1 and aimed to evaluate the women’s level of knowledge about screening and HPV vaccination and their access cervical-cancer-specific healthcare services.

### Statistical Analysis 

Data were collected and analyzed in Excel 2019. Data were labelled as nominal or quantitative variables. Nominal variables were characterized by means of frequencies.

## 3. Results

### 3.1. Burden of Cervical Cancer in the Bucharest Oncology Institute

By searching the electronic database of our hospital, we were faced with the huge burden that cervical cancer (CC) places on the healthcare system in Romania, both in terms of the number of cases and the severity of disease. Over the last decade, cervical cancer accounted for 17.97 to 29.56 percent of the total cases treated in the Bucharest Oncology Institute. A total of 15 to 30 percent of all surgical procedures performed in a year were related to malignant tumors of the cervix. Most cases (>87% each year) were diagnosed at advanced stages (>stage Ib) of the disease that required more than local treatment. Around 3.5–5.9% of the surgical CC cases treated each year required extreme surgical procedures (such as pelvic exenterations) to ensure optimal treatment for the patient. This, combined with the severity of the associated pathology of our patients (many being elderly or having cardiac, pulmonary, renal, or other chronic illnesses), led to an increase in the median duration of hospital stay for each episode and in the cost of treating the case. The median duration of hospital stay was 7.62 days in 2022, 10.12 days in 2021 and 10.25 days in 2020, with extremes varying from 1 to 56 days. The cost per CC case treated has increased constantly in recent years, thus further burdening a healthcare system already underfunded. The cost has more than doubled in the past 3 years, reaching a value of 12.149 RON/case treated in 2022 (vs. 5146 RON/case treated in 2020). This cost represents the total amount paid by the hospital for the medication, equipment, devices needed for treating a case and the costs associated with the patient’s hospital stay (electricity, water, cleaning services, food and so on). The cost does not include the salaries of the medical personal involved in the treatment of the case. This cost is not supported by the patient and is just a statistical analysis of the cost of hospitalization generated automatically by the database upon request. The numbers presented represent how much the hospital pays in order to treat a case. Each case treated is reported and the state (the primary health insurer for all citizens) refunds to the hospital a fixed amount calculated according to the index of the case complexity, which is fixed for any cervical cancer case regardless of additional costs to the hospital. At present, the state refunds to the hospital less than 7.000 RON/cervical cancer case treated. This refund value has not increased in recent years. Therefore, as a result of the current refunding system, each case treated for cervical cancer represents a significant financial deficit in the budget of the hospital, thus endangering the future care of these patients.

During the pandemic years (2020–2022), we saw a reduction in cervical cancer cases that was in parallel with the overall reduction in all cancer cases treated. This was caused by the restrictions imposed during the COVID-19 pandemic and the patients’ fear of contracting the SARS-CoV2 virus when being treated for other pathologies, which severely limited the access of patients to treatment and resulted in a huge impact on the treatment and oncologic outcomes for cancer patients [8]. However, the percentage of hospitalization episodes and number of surgical procedures caused by CC out of the total remained the same as during the pre-pandemic years. The restrictions were only lifted in 2022. The general characteristics of this pathology in our institute are summarized in Table 2.

### 3.2. Knowledge about Cervical Cancer Prevention, Early Detection Methods and Healthcare Access among Targeted Population

The second component of our study consisted of a cross-sectional study of the women participating in the National Screening Program for Cervical Cancer during Pap smear testing campaigns with the mobile screening unit that were organized in the counties of Tulcea, Buzau and Ilfov. A total of 528 women answered a 14-question questionnaire evaluating demographic data, their level of knowledge about screening and HPV vaccination and their access to cervical-cancer-specific healthcare services (Table 1).

#### 3.2.1. Demographic Data of Respondents

The general demographic characteristics of respondents are presented in Table 3. The median age of the respondents was 47.86 years, with extremes ranging from 19 to 69 years old. A total of 28.22% (149 women) were residents of urban areas and 71.78% (379 women) were residents of underprivileged rural areas in Romania. Two percent of our respondents (12 women) did not go to school ever and 13.85% only finished primary-level education (71 women). Most of our respondents (62.12%) had finished high school or at least went to 2 out of 4 years of high school (260 finished high school, 68 had 10th grade education). Only 20.83% had post-high-school education (48 had professional short-term superior degrees and 62 bachelor’s or master’s degrees). A total of 521 out of 528 respondents told us about their education.

#### 3.2.2. Access to Cervical-Cancer-Specific Preventive/Diagnostic Methods

As a measure of access to specific preventive methods, we asked the respondents if they had previously had a Pap smear. All participants answered the question. More than a fifth of our respondents (21%—112 women) had never had a Pap smear in their life. Of the 78.78% (416 women) of women that had previously had a Pap smear, 92.6% (384 women) recalled when they had it. A total of 77 women had been tested in the previous year, 137 women had a Pap smear 2 to 4 years before and for 170 women it had been 5 years or more since the test. Most women in this last category had only been tested once in their life (usually the test was performed before childbirth and not by choice). For 73 of these women, it had been more than 10 years since the test, whereas it was more than 20 years for 26 women.

We also asked our respondents if they would have gotten screened for cervical cancer by choice if the mobile screening unit did not come to their city; 526 women answered this question. Only 43.65% (233 women) answered yes, whereas 55.49% (293 women) had no intention of getting tested by choice/demand. A total of 13 women gave no reason for this and 15 gave multiple reasons. Each multiple reason was split into component reasons for statistical analysis.

The most frequent reasons given for the lack of intention to get a Pap smear in the absence of the immediate opportunity presented by the mobile screening unit were, in order of frequency: lack of time (45.33%), lack of financial means (30.66%) and the absence of any symptomatology (14%). Among other reasons given, we identify the lack of information about where/how to get tested, fear of the procedure/test results and the need to travel to another city in order to get tested.

Another question was aimed at evaluating the access of respondents to supplementary diagnostic testing specific for cervical cancer. We asked about the probability of undergoing supplementary tests if needed and the reasons for not attending those tests. A total of 458 women responded to this question (86.74%). Most women (94.54%) responded that they will definitely/probably go to supplementary diagnostic tests if the Pap smear results indicated the need. However, 4.8% responded they would definitely/probably not go due to lack of financial means, indicating an acute lack of knowledge about the possibility of having those tests free of charge in state hospitals. Table 4 summarizes the access to cervical-cancer-specific preventive/diagnostic methods responses of the respondents in our group.

#### 3.2.3. Knowledge about HPV, HPV Vaccination and Access to Vaccine

Regarding HPV infection, most of the respondents had heard of this infection and were aware of the link between HPV and cervical cancer. A total of 524 women answered this question and only 11% (60 women) did not know about HPV and its link to cervical malignant tumors. During our analysis, we saw that even though most women who do not know about the connection between HPV infection and cervical cancer have no or only primary education, they are not the only ones, women with superior education accounted for 5% (3 women with “No” responses).

However, although most respondents (79%—416 out of 528 respondents) are aware there is an effective vaccine against HPV, most (96%—503 out of 524 respondents) are not vaccinated. Only 22 women out of 524 respondents declared that they were HPV vaccinated. Vaccinated women have a maximum age of 54, almost half live in urban areas and all have high-school education or higher. All women with higher education know that there is an effective HPV vaccine. In our group, 26 women were under the age of 30 and would have been eligible for the national HPV vaccination program. Out of those 26, only 4 were vaccinated (representing a 15.38% vaccination uptake in the eligible population of the respondent group). The remaining 22 vaccinated women did so on their own volition and supporting the full cost of the vaccine. This is an extremely low vaccination rate through a national vaccination program, especially when we compare it with other European countries, in which vaccination rates tend to be over 40% of the eligible population [9].

As a final analysis, we tried to evaluate the knowledge of our respondents about the existence of the national HPV vaccination program that offers free vaccination for girls aged 11 to 18 and their intent for letting their daughters participate. Only 57% of the women knew about this possibility (295 of 521 respondents). Even after being told about the possibility of free HPV vaccination, only 57% of the women would vaccinate their daughters (271 out of 471 respondents). Table 5 summarizes the knowledge about HPV, HPV vaccination and access to the vaccine in the study group.

## 4. Discussion

### 4.1. Cervical Cancer in Romania vs. Europe/World—Current Situation, Trends, Incidence and Mortality

Romania is a country located in the southeastern part of Europe, it has been a member of the European Union (EU) since 2007 and had a population of 19.04 million resident citizens in 2022 according to the National Statistics Institute, 9.80 million of them were women [10]. Romania has the highest rural population (in terms of proportion of total population) among EU member states, with a value of 46% [11]. Life expectancy for women in Romania is 79.3 years at birth, below the EU average [12]. Cervical cancer represents the fifth highest cancer-related cause of death in Romania, after lung, breast, colon, and prostate [10]. For most cancers, Romania follows the European trends, but for cervical cancer the country’s situation is extremely alarming, with high rates of incidence and mortality. In Figure 1 we present the country’s situation in comparison to the European average for the top five female cancers using data from GLOBOCAN 2020 and the Human Papillomavirus and related Disease Report WORLD [2,13]. Even so, the data for Romania may be underestimated, since the country does not have a functional cancer registry for the accurate monitoring of incidence or mortality rates, nor does Romania have a registry for recording a woman’s individual screening history [14], which would be helpful in evaluating potential preventive measures. Additionally, testing laboratories, hospitals and doctors are not required to report screening results and there is no possible way of correctly evaluating the screening coverage among the targeted age group.

For 2020, the European Union, through the European Cancer Information System (ECIS) [15], has presented relevant information on cervical cancer incidence and mortality rates (expressed as age-standardized rated or ASR per 100,000 women) for all 27 member states of the EU, thus allowing us to compare the rates in Romania with the other member states and the European average. Using data freely available on ECIS [15], we generated Figure 2, Figure 3 and Figure 4, depicting the cervical cancer situation in Romania in 2020, the reduction in the incidence and mortality in 2020 compared with 2015 and also a projection of cervical cancer in 2040 based on current data for all member states of EU 27 (European Union—27 member states format, after 2020 when the United Kingdom retired from EU). From these figures, we can see that despite having the largest reduction between 2015 and 2020 in both incidence and mortality rates for CC among the EU 27 states, Romania still has the highest incidence and mortality rates of the EU 27 countries. The projections for 2040 show that Romania is likely to have one of the most significant reductions in both incidence and mortality rates for cervical cancer among the EU 27 states, and we believe this is partly due to the significant improvement that Romania’s CC-specific preventive measures may have in the context of the EU directive demanding certain goals for the CC eradication program.

Romania, like other European Union member states, has several serious public health issues that still need to be addressed. However, significant contrasts are observed between older and newer member states when it comes to incidence and mortality rates for cervical cancer. These inequities between states might be explained by differences in the preventive strategies adopted by each country during the time [16] and the duration for which those strategies were in effect.

Since December 2003, there has been an EU Council recommendation addressed to member states to implement population-based systematic screening programs for cervical, breast and colorectal cancers [17]. In 2017, 22 out of 28 member states had implemented a form of cervical cancer screening program (national or regional), ensuring an average coverage of 59.2% of women between 30 to 59 years of age, with a participation rate of 50.7% [18]. The screening programs developed by member states varied widely in both the targeted age range, methods used for screening, form of screening (voluntary/compulsory/invitation based) and coverage of certain regions (discrepancies between rural and urban areas or covering only some regions as pilot programs) [16,19]. Given the heterogeneity between various national programs, it is hard to compare the quality of testing, the monitoring and evaluation methods or the cost effectiveness of each program [18,20,21]. However, there is a clear connection between access to organized screening programs and cervical-cancer-specific incidence and mortality rates [18,22,23,24].

The previous studies [16] that proved the positive impact that functional screening programs had in various EU states, also showed that Romania is in a situation similar to that of other EU states in the 1960s–1970s, before the implementation of national screening programs. In those countries, the cervical-cancer-specific mortality rates decreased significantly after the initiation of screening programs [25]. According to statistical research conducted by Eurostat in 2022 comparing the self-reported last cervical smear test among women aged 20 to 69 in EU countries, Romania placed second in the number of women that have never been tested, giving us an image closer to reality about the participation in routine screening for Romanian women (47.4% of them never had a Pap smear, compared with the EU average of 13.4%) [26]—Figure 5.

A functional national screening program is essential for early diagnosis and potentially curative therapeutic intervention [27]. Studies have shown that even a single screening in a lifetime can significantly decrease the mortality and incidence of advanced cervical cancer compared with no screening [28]. Some studies have shown that women attending screening programs benefit by a 41 to 92% reduction in cervical cancer mortality [29]. Cervical-cancer-specific mortality rates have decreased over recent years in both Romania and the EU (there is a 56% reduction for Romania and a 58% reduction for the EU when we compare the cervical cancer mortality rates in 2016 vs. 2002). However, when directly comparing the mortality rates of Romania vs. EU, we see a huge difference (in 2002 the mortality rate difference was 276%, and in 2016 the gap remained the same—277%) [30]. Among Romania’s regions, we see that a huge gap between mortality rates in rural and urban areas persists, despite an impressive 44% reduction in mortality over the past two decades in some rural areas situated in the northwestern and western parts of the country. The rural–urban mortality gap was 24% in 2019 [30].

### 4.2. Cervical Cancer Screening Program in Romania

Screening for cervical cancer in Romania started with a population-based pilot program conducted in Cluj County between 2002 and 2008. The coverage of this regional pilot program was 21% by the end of 2008 [19]. In 2012, the state started a 5-year nationwide pilot screening program for cervical cancer [31]. As a design, the program consisted of establishing a network of medical healthcare workers that had a target of testing all eligible women aged 25 to 64 years old by Pap smear once every 5 years, regardless of insurance status and with no symptomatology and no previous cervical cancer diagnosis or total hysterectomy. The eligible population was evaluated at over 6 million women in across the whole country. General practitioners and obstetrics–gynecology specialists were forced to participate by law. At the end of the pilot program, 730.000 women were tested (13.6% of eligible women) [32]. However, the program had limited success (at its peak in 2014, the national coverage of eligible women was 8%). The reason for this failure was the weak participation of doctors, with only 48% of them actively recommending testing through the program. The main reason cited was the complicated procedures and documents [1]. At the same time, the funds allocated to the program were small and infrequent, which meant that often a woman went to the gynecologist to be tested but could not be tested due to lack of funds. This, as well as the delay in providing the results, seriously affected the credibility of the program. Another reason for the failure of the pilot program was an almost non-existent coverage in rural areas [1].

After the experience gained through this pilot program, several measures were implemented to remedy the identified shortcomings, namely: (1) regional monitoring and management units of the program were established that reduced the problems related to the delay of results, bureaucracy and procedures, ensuring at the same time the external control of the quality of the testing procedures and the issued results; and (2) the possibility of covering rural regions was created through the operationalization of mobile screening units.

However, even in the years 2018–2023 the program did not register the expected success for the following reasons: (1) most doctors believe that the program no longer works; (2) most eligible women do not know about the possibility of free Babes–Papanicolaou testing through the program; (3) very few medical personnel are qualified, according to the methodological norms, to collect the tests (only gynecologists and general practitioners [33]), which makes it very difficult to find doctors willing to go on screening campaigns; and (4) the lack of coherent funding causing a delay of 5–6 months in the payment of the services performed, causes the discouragement of the staff and ultimately the abandonment of the program.

In 2020, the number of eligible women in Romania was, according to the National Statistics Institute, 5.1 million. The same institute provides us with data about the coverage of the eligible women by the national screening program. The percentage of eligible women covered by testing in the previous 5 years was 2.6% in 2018, 3.1% in 2019 and 3.4% in 2020 (2020 is the last year for which there is data available; however, the data is partial and still being compiled by the institute) [34]. This translates into roughly 175,000 women having been tested in the last 5 years. However, according to Romania’s cervical cancer country profile [35], 38% of women of eligible age report they have been screened in the last 5 years. This demonstrates that most of the cervical cancer screening is opportunistic rather than systematic, contradicting the EU [17] and WHO (World Health Organization) recommendations on this matter.

The proportion of women with a high level of education that are screened is almost five times higher than that for women with a low level of education. In terms of income categories, the proportion of women with high incomes who benefit from a screening test is three times higher than that of women with low incomes (43.3% vs. 12.8%) [25]. In Romania, more than 10% of the population consists of various minorities [34], and 3.3% belong to the Roma community, although this number may be underestimated [36]. It is a well-established fact that non-ethnic Romanians, especially Roma women, participate less frequently in screening programs compared with the main ethnic group [37,38]. This aspect is linked to a lower level of knowledge about the existence of screening programs and mistrust about the free of charge policy of the screening programs and about the benefits screening could bring [14,38]. Anterior cross-sectional studies using questionnaires addressed to both Romanian and ethnic minority populations in Romania showed that the low attendance in screening procedures is caused by a lack of awareness about the existence of the screening program [1] and a perceived lack of financial means to get screened and follow up positive test results [1,39]. Although Romanian women have a positive attitude towards preventive medicine [40], this does not necessarily correlate with actual participation in screening programs, mostly because the perceived subjective “costs” of having a test is significantly higher than the perceived benefits; this is especially true for women who have never had a Pap smear [41].

### 4.3. HPV Vaccination Program in Romania

In Romania, there is also a national program that provides a free anti-HPV vaccine through the family doctor. HPV vaccination is an extremely controversial subject in Romania and is not mandatory nor covered by the National Vaccination Program [42]. The HPV vaccination program was introduced for the first time in 2008, when the vaccine was offered for 10–11-year-old girls (only 2.57% of the total eligible population was vaccinated then). In 2009, the eligibility age was extended to 14 years; however, in 2010, the program was discontinued due to lack of participation. Starting from January 2021, the program was resumed for girls aged 11–14 in a modified form, and from September 2021, the eligible age was extended to 18 years. Therefore, HPV vaccination in Romania is currently free upon request at the family doctor for girls aged 11–18. The rest of the female population and male adolescents are not eligible for free vaccination through the program; these categories are only able to get vaccinated by supporting he full cost. The full cost of three doses of the HPV vaccine Gardasil-9 is the equivalent of the current in-pay minimum wage. The Ministry of Health, although recommending the vaccination for boys, does not provide gender-neutral accessibility to vaccine, even though previous studies proved the benefit of the HPV vaccine even in men (especially in its nonavalent form) for preventing HPV-associated diseases [43] and reducing low- and high-grade cervical intraepithelial lesions [44].

Like the national screening program, the national HPV vaccination program is also a failure (Romania has an extremely low rate of HPV vaccination, only 12–13% of the eligible population [45]); however, in this case the main causes are more related to an acute lack of information for the parents and eligible girls [46,47] or to the parents obtaining information from dubious sources (social media, internet, mass media, etc.) [48], as well as for reasons related to the parents’ beliefs, attitudes and social prejudices. Poor participation in all vaccination campaigns was generated by deficient public health strategies—there were enough vaccine doses to ensure coverage of the targeted population, there was adequate diffusion in all areas and in locations that facilitated access (the vaccination campaigns were implemented in schools) and there were enough doctors to administer the vaccine [49]. The only reason for the failure of the vaccination programs remains the lack of adequate information campaigns addressing the lack of knowledge about the connection between HPV and cancer and the general lack of acknowledgement of the importance and benefits of HPV vaccination.

Given the fact that, in Romania, parental consent and an active request for the HPV vaccine is required for children under 18 years to be vaccinated, it is paramount to understand parents’ profiles, beliefs and general acceptance of the vaccine in order to optimize vaccination and improve participation rates. The facilitators for parents to be willing to vaccinate their children against HPV include obtaining information from medical sources [49], female gender [49,50], urban settings [51,52], higher income status and higher educational level [53,54]. As with other vaccines, the physician’s recommendations sway parental consent [55]. Hesitance regarding vaccination is generated by a preconceived notion from the parents that HPV vaccination may adversely influence the sexual behavior of their children [56] or conspiracy beliefs stimulated by the mass media or social platforms [48]. The other most common reasons for unwillingness to vaccinate are doubts about the efficacy of vaccine, the cost and fear of side effects [56,57].

Human papilloma virus is the most common sexually transmitted viral infection in the world [58,59]; most sexually active people come across at least one genotype of the virus during their lifetime [47]. Chronic HPV infection can lead to head and neck [60,61,62,63,64], skin [65,66,67,68] or anogenital [69,70] cancers, causes enormous medical costs [68] and can severely affect fertility in both men and women [71,72,73] and alter the efficacy of assisted-reproduction techniques [74,75]. We can see that HPV infection seriously affects both genders, and it becomes apparent that, in order to meet the World Health Organization’s goal of eradicating cervical cancer, a gender-neutral HPV vaccination strategy needs to be applied; this is, even with a moderate coverage, the most effective way of achieving this goal [76,77,78]. A gender-neutral campaign also favors the development of a herd immunization effect among adolescents, which will have beneficial results [79].

In conclusion, all future HPV vaccination programs in Romania need to consider the lack of information and confusion regarding adverse effects, which can be solved through education through intense information campaigns. The HPV vaccination programs need to be gender-neutral in order to achieve optimum herd immunization effect and need to be fully state funded in order to address the accessibility issues that are real or perceived.

### 4.4. Other Deterrents of Attendance for Screening and HPV Vaccine Uptake in Romania

One of the main reasons for the dimensions of this pathology in Romania is the amount that the country allocates for disease prevention from the total healthcare expenditure. In 2015, Romania allocated a little over 2% of the health budget for the prevention of diseases (this represents the total budget allocated for all preventive measures for all diseases and not just for the specific methods for the prevention of cervical cancer). With the exception of for 2015, the allocated percentage varied between 1.18 and 1.76%. This is data freely available on the National Statistical Institute site. The data are available for years up to 2019 [10].

Other causes of the failure of the preventive measures in Romania are the health illiteracy of the population and the general functional illiteracy. In Romania, access to education is guaranteed and non-discriminatory. Going to school has been mandatory and free for both primary and medium levels (for a total of 12 or 13 years) since 2020. Even before 2020, school was mandatory and free for eight grades, which meant finishing the primary cycle. Parents can be fined for not sending their children to school, and many non-profit and charitable organizations are helping children to be able to go to school. However, we saw in our study group that 2.27% of respondents never went to school, were unable to read and needed assistance in completing the questionnaire. The level of functional illiteracy or low functionality among the Romanian population is extremely high (around 89% according to some statistics) [80]. Functional illiteracy is defined as the lack of the ability to interpret and integrate ideas and information from a given text, deduce cause–effect relationships and hypotheses, establish similarities and differences between characters, facts, events or concepts and draw an overall conclusion from the text. As proof of this, we encountered respondents that, even though they said they did not know about the existence of the HPV vaccine, answered “YES” to the question “Are you HPV vaccinated?”. Several studies have proved the correlation between a low level of literacy and low level of knowledge regarding health services and health outcomes [81,82,83,84,85], low participation in screening measures [82,83], low adherence to diagnostic and treatment regimens [85], poorer health outcomes [86,87,88] and health behaviors [89,90,91,92] and increased hospitalizations and healthcare-associated costs [93,94].

All these reasons are important, but the fact that most people perceive that they need financial means in order to get tested, diagnosed or treated and do not know where to get specific medical services is truly confounding. This comes as a surprise, since in Romania access to healthcare is guaranteed by the constitution, the healthcare system is based on equality of accessibility to medical services regardless of financial contribution of each citizen and many underprivileged citizens are freely insured for healthcare. The state-funded system provides free access to preventive medical services; diagnostic procedures and investigations (including complex imaging that can help in differentiating between metastatic lesions and primary/other lesions [95,96] or diagnostic synchronous lesions [97]); all treatment options for both precursor lesions and for invasive cancer and simultaneous treatment of primary and metastatic lesions [98] through national treatment programs [99] (surgery—including novel techniques such as laparoscopy for both oncologic diagnosis and staging, radical or palliative treatment [100], robotic surgery, novel surgical techniques for cancer [101], sentinel lymph node identification with radioactive material, which has become the standard of care for some cancers [102], or indocyanine green [103], reconstructive surgery, identification of occult cancer lesions using radioactive colloids [104], radiotherapy, chemotherapy, immunotherapy [105], personalized oncological treatments and so on); and also for follow-up procedures and investigations. All being said, it becomes apparent that focused information campaigns are of paramount importance for resolving the problems that cervical cancer is causing in Romania, as health illiteracy is the primary reason for the current situation.

## 5. Limitations

The main limitation of our study is the fact that mobile unit screening campaigns for cervical cancer are usually addressed to women in rural areas who have limited access to healthcare and lower education levels. Our results are surely influenced by this factor, and we believe that the results may change if the questionnaire was answered by a population consisting of both women in underprivileged areas and women in high-density urban areas. Another limitation of our study is the functional illiteracy present in the study group, which means that even if some of our respondents declare they went to school and they can read, some of them do not understand the questions and prefer not to answer or answer by guessing, thus affecting our results.

## 6. Conclusions

The responses of our patients indicate the marked need for sustained population information campaigns (such as those conducted in Romania for breast cancer) highlighting the impact, symptomatology, and preventive and diagnostic methods associated with cervical cancer, the association of HPV infection with cervical cancer and the existence of the national screening and vaccination programs that offer free screening and access to vaccination.

For the success of national programs aiming to reduce the impact of cervical cancer, the Ministry of Health should organize coherent and sustained information campaigns addressed to both patients and doctors, should ensure the continuity of the funds intended for the programs (thus avoiding delays in the payment of services), simplify the bureaucratic procedures and modify the functioning of the screening program by allowing the participation of other medical categories, thus avoiding staff shortages. Another important factor that could help in mitigating the impact of cervical cancer in Romania is the gender neutrality of HPV vaccination campaigns.

## Figures and Tables

**Figure 1 diagnostics-13-02776-f001:**
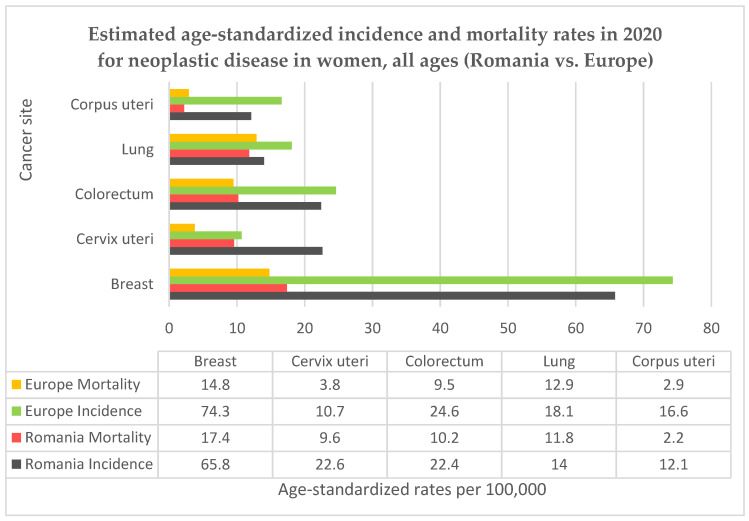
Comparison between Romanian and European incidence and mortality rates for neoplastic diseases in women.

**Figure 2 diagnostics-13-02776-f002:**
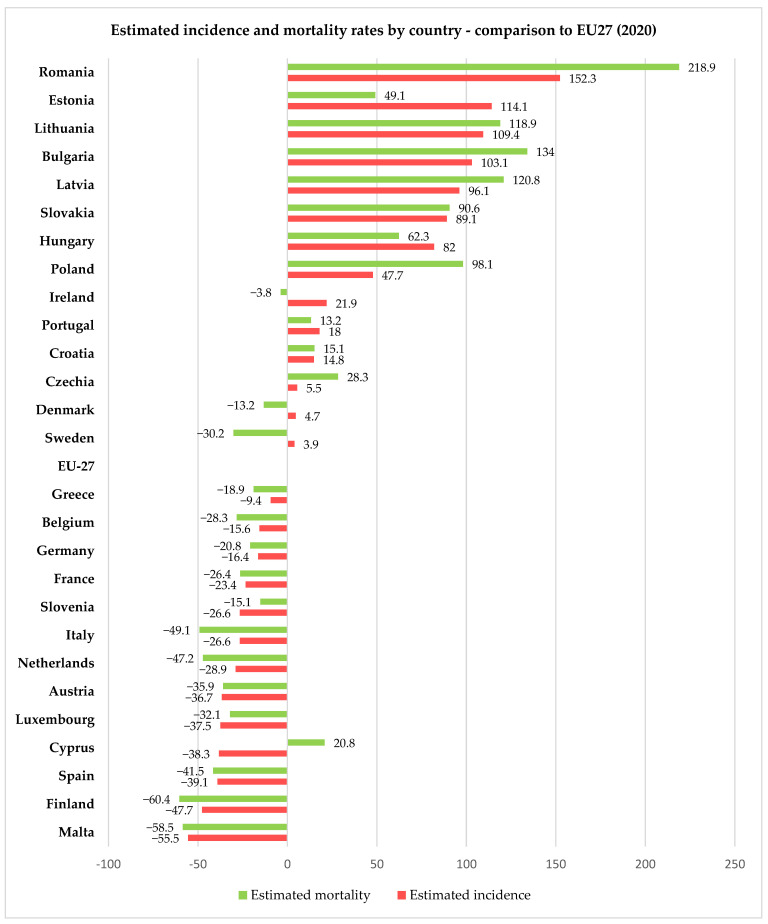
Comparison between Romanian and European incidence and mortality rates (expressed as ASR/100,000) for women with cervical cancer in 2020 according to ECIS [15].

**Figure 3 diagnostics-13-02776-f003:**
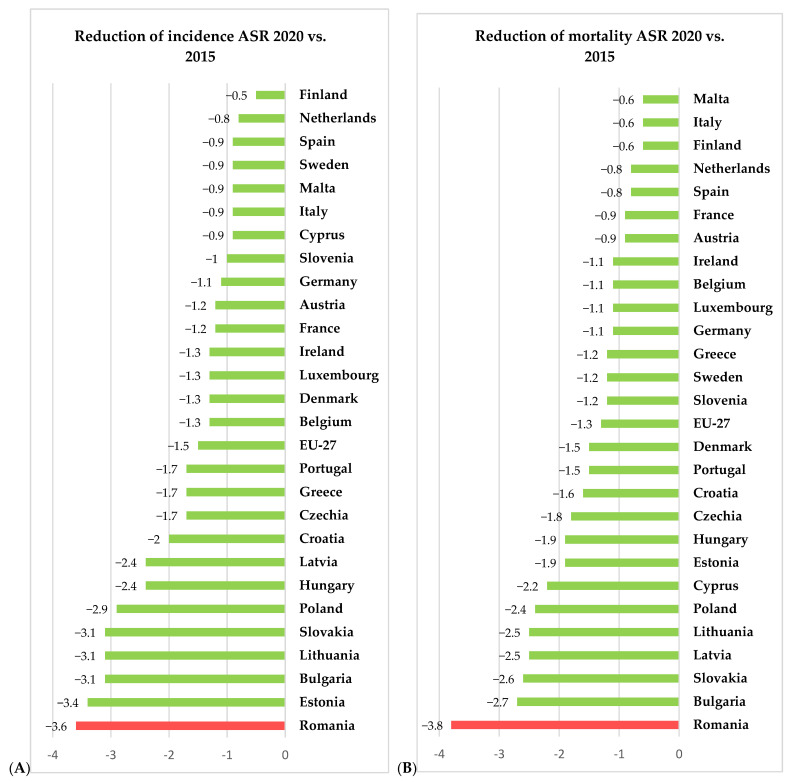
Situation of incidence (**A**) and mortality (**B**) ASR (expressed as ASR/100,000) for cervical cancer for 2020 vs. 2015—comparison between Romanian and EU27 countries according to ECIS [15].

**Figure 4 diagnostics-13-02776-f004:**
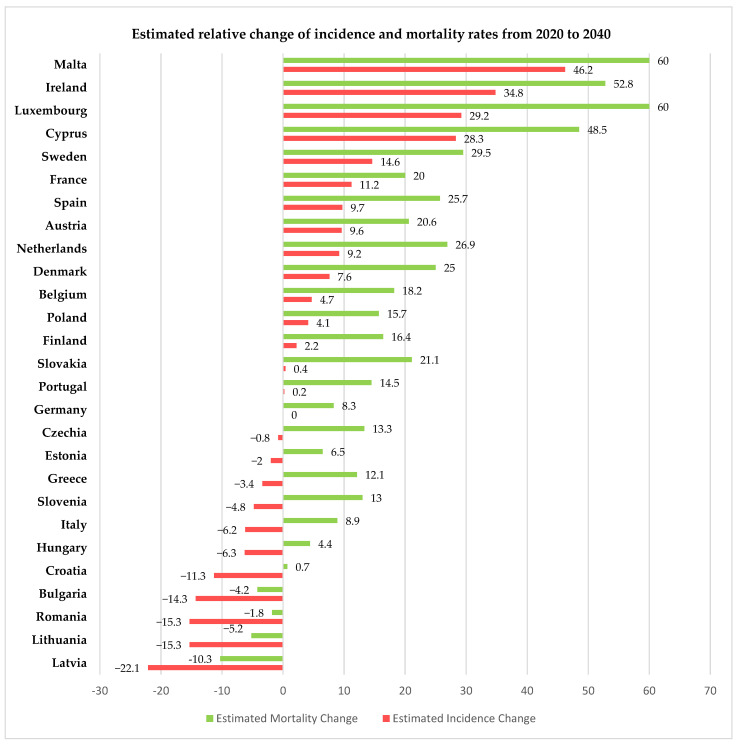
Estimated change in incidence and mortality rates in EU 27 countries for 2020 vs. 2040 according to ECIS [15].

**Figure 5 diagnostics-13-02776-f005:**
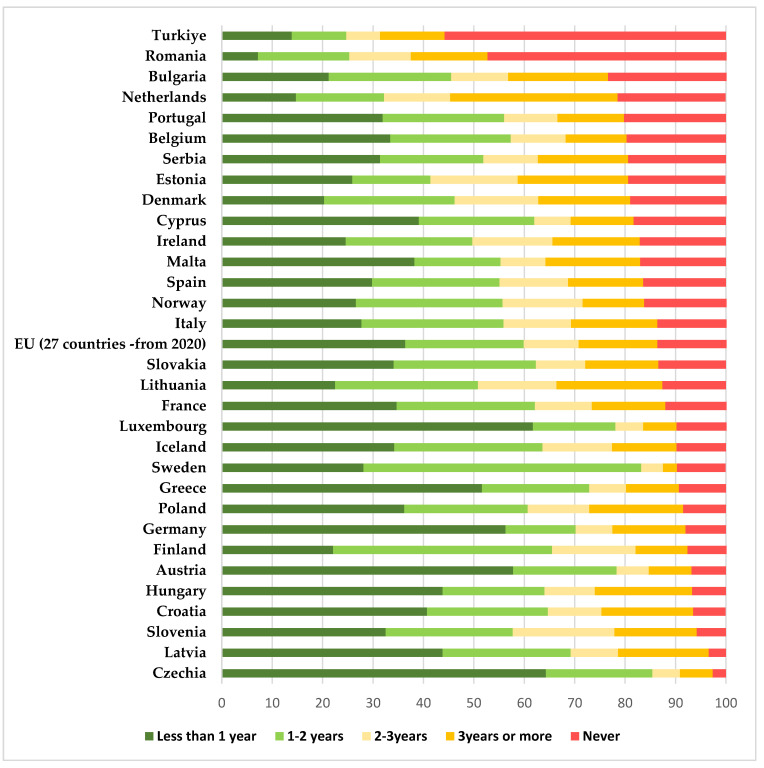
Eurostat 2022 statistics comparing the self-reported last cervical smear test among women aged 20 to 69 in EU countries (results are expressed as percentages)—based on the Eurostat results freely available online [26].

**Table 1 diagnostics-13-02776-t001:** Characteristics of questionnaire applied to the women participating in the National Screening Program for Cervical Cancer during testing campaigns in disadvantaged parts of Romania.

	Question	Possible Responses
1	Level of education	No education 8th grade education 10th grade education High-school degree/ Professional degree Bachelor’s degree or higher
2	Age	Numerical
3	Place of residence	Name of town/village
4	Have you ever been tested before by Pap smear?	YES NO
5	When were you tested by Pap smear?	Free-form answer
6	Would you have gone to be tested if the mobile screening unit did not come to your town/village?	YES NO
7	If the answer to the previous question is NO then answer why not?	Lack of financial means Lack of time I do not know where to get tested Nothing is bothering me Other reasons
8	If you will get a test result that requires other investigations, will you do them?	Yes Probably yes Probably no, due to the lack of financial means Probably no, due to other reasons No
9	Have you ever heard that HPV infection can lead to cervical cancer?	YES NO
10	Do you know there is an HPV vaccine available?	YES NO
11	Are you HPV vaccinated?	YES NO
12	Are you aware that the HPV vaccine is freely available for girls at the general practitioner?	YES NO
13	Knowing this, would you vaccinate your daughters?	YES NO
14	If the answer to the previous question is NO, then answer why not?	Free-form answer

Some answers were recorded in order to be statistically analyzed.

**Table 2 diagnostics-13-02776-t002:** Disease burden associated with cervical cancer (CC) in the Bucharest Oncology Institute 2012–2022: General overview.

	2022	2021	2020	2019	2018	2017	2016	2015	2014	2013	2012
Hospitalization episodes for CC Number (% of total no. of hospitalization episodes—all pathologies)	3726 (29.56)	3028 (27.62)	2460 (21.14)	3775 (17.97)	4590 (21.52)	4679 (22.12)	4585 (20.79)	4315 (19.27)	5648 (24.92)	5734 (26.03)	4906 (22.61)
No. of patients treated for CC	967	827	805	1290	1456	1543	1620	1585	1918	1994	1817
Surgical procedures for CC	668	651	642	1204	1251	1180	1274	1155	1911	1925	1868
Number (% of total no. of surgical procedures performed—all pathologies)	(15.09)	(16.49)	(18.28)	(18.99)	(19.28)	(18.46)	(19.5)	(17.47)	(29.18)	(29.11)	(28.8)
Advanced-stage CC (%)	92.15	94.61	87.32	89.14	91.57	90.86	91.79	89.99	93.63	92.47	91.52

**Table 3 diagnostics-13-02776-t003:** Demographic characteristics of the respondents.

Demographic Characteristics of Respondents			
	Category	Number	Percentage (%)
Age	19–24 years	5	0.95
	25–29 years	20	3.79
	30–34 years	35	6.63
	35–39 years	39	7.39
	40–44 years	72	13.64
	45–49 years	105	19.88
	50–54 years	125	23.67
	55–59 years	75	14.20
	60–64 years	47	8.90
	65–69 years	5	0.95
Level of education	No education	12	2.27
	8th grade education	71	13.45
	10th grade education	68	12.88
	High-school degree or professional degree	260	49.24
	Bachelor’s degree or higher	110	20.83
	NA	7	1.33
Place of residence	Urban	149	28.22
	Rural	379	71.78

**Table 4 diagnostics-13-02776-t004:** Access to cervical cancer specific preventive/diagnostic methods.

Access to Cervical Cancer Specific Preventive/Diagnostic Methods			Number	Percentage(%)
(%) Previous Pap smear	YES		304 (#)	57.57
	NO		112	21.21
	NR		112	21.21
Time since last Pap smear	Less than 1 year		77	14.58
	2–4 years		137	25.94
	5–9 years		97	18.37
	10–19 years		47	8.90
	20+ years		26	4.92
	NR		144	27.27
Intention of getting a routine Pap smear if not readily available with the mobile screening unit	YES		233	43.65
	NO		293	55.49
	NR		2	0.37
Reasons for lack of intention for getting a routine Pap smear	Lack of financial means		92	17.42
	Lack of time		136	25.75
	I do not know where to get tested		16	3.03
	Nothing is bothering me		42	7.95
	Fear of procedure/results		6	1.13
	Other reasons		8	1.51
	NR		228	43.18
Intention of going for additional investigations if the results of the Pap smear indicate the need	Yes		334	63.25
	Probably yes		99	18.75
	Probably/definitely no	Lack of financial means	22	4.16
		Other reasons	3	0.56
	NR		70	13.25

NR—no response; # 227 respondents had only had one Pap smear during their lifetime, conducted before childbirth and not by choice.

**Table 5 diagnostics-13-02776-t005:** Knowledge about HPV, HPV vaccination and access to vaccine.

Knowledge about HPV, HPV Vaccination and Access to Vaccine		Number
Knowledge about the connection between HPV and cervical cancer	YES	464
	NO	60
	NR	4
Knowledge about the existence of an HPV vaccine	YES	416
	NO	112
	NR	0
Women eligible during lifetime for free HPV vaccination through national vaccination program	Total number	22
	Vaccinated through the national program	4
	Not vaccinated	18
Uptake of HPV vaccination	YES	21
	NO	503
	NR	4
Knowledge about the free HPV vaccination national program	YES	295
	NO	226
	NR	7
Respondents’ intentions for HPV vaccination of their daughters	YES	271
	NO	208
	NR	49
Reasons given for not wanting to vaccinate children (#)	Lack of information on vaccine	33
	Lack of information on free vaccination program	7
	Lack of trust in the vaccine	7
	Vaccine was not available	9
	Not having children	17
	Believing vaccination should be addressed to girls over 18 years old (fear of modifying sexual behavior of child)	3
	Fear of side effects	7
	Desire of letting the child choose if she gets vaccinated or not	10
	Child is under the target age	2
	NR	431

NR—no response; # 97 respondents gave a reason for not wanting to vaccinate children (out of 208 women who would not HPV vaccinate children).

## Data Availability

Since it involves personal data, due to privacy issues, data will be available upon request by e-mail to E.C.

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
