# Peer review of "Inequities in Screening and HPV Vaccination Programs and Their Impact on Cervical Cancer Statistics in Romania"

_diagnostics, 2023, doi:10.3390/diagnostics13172776_

Round 1

Reviewer 1 Report

The article deals with an important topic, that is, the low uptake of cervical cancer prevention measures  in Romania and high incidence and mortality. Unfortunately, the manuscript did not markedly improve my insight into the problem in Romania, partly due to its methodological quality which is fair at best.

Some general comments are:

1. The inclusion of the first component of the study (i.e. descriptive statistics of internal electronic database) is not clear in light of the goal of the study. The authors report on the incidence/mortality/costs/number of hospital days  based on data collected within an academic Bucharest hospital. The data are presented with very little context. The costs are not further specified and seem to be tariffs charged to the health insurer or patient. Table 2 is difficult to read and its layout should be improved.

2. The questionnaire is more interesting and I would personally favor a paper with only the questionnaire results. The questionnaire study however has serious limitations. First, the questionnaire results on screening are presented as rather extreme and the authors indicate that many study participants have a very poor screening history. This is not true. Previous screening uptake in this population is similar similar to those observed in other  European countries. For instance, 56% of the participants had a Pap smear 0-4 years ago, which seems acceptable. Therefore, because of the high cancer incidence in Romania, I tend to conclude that the study population of women participating in screening via the mobile unit is not representative of the Romanian screen-eligible population. Second, the authors indicate that only 4% of the participants are vaccinated. This is not a surprising result because the median age was 46 years which means that most women in the study are not eligible for prophylactic vaccination. The large majority of women in the study also know about the vaccine and about HPV, which is in contrast with the Romanian population which, according to the authors, have an extreme health illiteracy (line 61). Third, 57% of the mothers indicated that they would like to have their daughters vaccinated. This is not an uncommon figure in countries with moderate vaccine uptake and not in line with the extremely low uptake in Romania.

The introduction is not written in a very scientifically, objective way, it reads more as a plea for action, illustrated for instance by a phrase like “the question arises how Romania ended up in this situation”. Please describe the situation in a more neutral way. Also do not mention specific names of other countries (l.50-51), but preferably use WHO classification LIC-LMIC-UMIC-HIC.

The results section contains many grammatical errors.

The pie charts are not attractive, there are much better ways to present the data.

Author Response

Dear Reviewer 1,

First, we the authors would like to express how deeply honored we are that our work was submitted to such a rigorous and extremely patient reviewer. We are delighted that you have dedicated your time to an in-depth analysis of our manuscript and we thank you for your suggestions which will allow us to improve our work and produce relevant material. After reviewing our material in accordance with your suggestions we also found a few additional small mistakes that we were able to rectify in time, and we are grateful for your contribution.

Second, we have addressed each of your concerns as follows:

  1. We revised the introduction as per your instructions. It is true this matter concerns us greatly and we tend to be a bit passionate about the subject. We rewritten it in a more neutral way. We removed the names of specific countries and used the terminology you suggested (WHO classification LIC-LMIC-UMIC-HIC)
  2. Grammatical and English errors - The document was subjected to multiple grammar and spelling checks and revised by one of our authors which has a Native Speaking Level English Language degree. Furthermore, if deemed necessary, we will use MDPI professional checking of our manuscript before printing.
  3. We removed the pie charts and presented the data as a Tables. We modified the text accordingly.
  4. About the inclusion of the first component of the study (i.e., descriptive statistics of internal electronic database), we revised this part of the manuscript and tried to better explain the aim of such query of our internal database. The Bucharest Institute of Oncology “Prof. Dr. Alexandru Trestioreanu” is one of 3 major regional cancer care facilities in Romania and is the primary facility for cancer care in the southern part of Romania (about a third of the entire Romanian territory and a population of 8 million out of Romania’s 20 million residents). Given this fact, the analysis of the pathology treated here, has the potential to high-light the burden placed by cervical cancer on Romania’s healthcare system. We added the explanation in Material and Methods section of the manuscript. We also added a section in Results - Burden of cervical cancer in Bucharest Oncology Institute better explaining what the costs represent and how the refund system works. Table 2 was revised in order to make it easier to read and its layout has been improved.
  5. We could not remove the internal database information because we consider it important in order to illustrate the burden of cervical cancer on the healthcare system in Romania. Also, in the opinion of the second reviewer, the database information was considered the interesting part of the article and he/she would have preferred an article containing only that information and not the ones acquired via the questionnaire. Since this is a personal preference of the reviewers, we opted for keeping both parts and trying to accommodate each reviewers’ concerns.
  6. We agree that the study population of women participating in screening via the mobile unit is not representative of the Romanian screen-eligible population, and we have acknowledged this as a limitation of our study in Section 5 – Limitations, rows 431-442 of unrevised manuscript and suggested that further studies on a more representative population may change the results. Furthermore, in Section 5 – Limitations we have discussed other limitations (like functional illiteracy) and why we believe some of the answers need to be considered carefully (with specific examples).
  7. We did not try to make a qualitative determination of the HPV vaccination rate since that would have been impossible given the limitations stated before. The 4% vaccination rate is not surprising at all and expresses the reality of our cohort. In our respondent group 26 women are under the age of 30 and would have been eligible for the national HPV vaccination program during their life-time. Out of those 26, only 4 were vaccinated (representing 15.38% vaccination uptake in eligible population of the responder’s group). The rest of 22 vaccinated women did so on their own volition and supporting the full cost of the vaccine. This 15.38% rate is an extremely low vaccination rate through a national vaccination program when we compared to other European countries in which vaccination rates tend to be over 40% of eligible population. We better explained this in the results section of the manuscript.
  8. We did not state that a 57% rate of the mothers willing to HPV-vaccinate their daughters is low. This again is a result in our cohort and is presented as such with no qualitative inferences. This rate is well within the medium for European countries for which there are data available, and is not surprising given the overall vaccine hesitancy we face in recent years.
  9. We indicated that many study participants have a very poor screening history and this is true. Please consider that 113 women did not answer this question and from the 416 women that answered only 384 recalled when they had the test. In our opinion this reflects the fact that they had an extremely long period since the test. For Figure 4. - Time since last Pap-smear, we only used data available from the 384 responders who previously had a Pap-smear and who record when they had it, so the 56% rate of women tested in the last 0-4 years does not reflect the reality of the whole group. In fact, a fifth of the group has never been tested and 32,8% did not recall when they had the test. When we consider the entire group of 528 participant then the rate of women who had a Pap-smear in the previous 0-4 years becomes 40.45%. We removed Figure 4. and presented the data as a table.
  10. We provided additional context in the Discussion section.

Reviewer 2 Report

Title of Manuscript: Inequities in Cervical Cancer Screening and HPV Vaccination Programs and Their Impact on Incidence/Mortality Rates and The Severity of Disease in Romania

GENERAL COMMENT: The manuscript addresses the urgent need to depict the reasons behind the high burden of cervical cancer on Romanian women, despite the existence of national cancer screening and HPV vaccination programs. What emerges as the cause of the extremely low number of screened and vaccinated women is a combination of factors: on the one hand a lack of information, time, and financial resources of the population, and on the other hand the underfunding and complicated access procedures of national screening and vaccination programs. While it seems that the authors identify what should be changed in their country by using a valid methodological approach, the general form of the paper could be significantly improved. For instance, results could be described in a clearer way, the introduction could be extended, and the discussion could be simplified.

Moreover, the title appears to be too long and hard to read, it would be appropriate to simplify and shorten it.

INTRODUCTION:

This section appears to be extremely short. It could be extended by placing more information about, for example, HPV-related diseases, HPV vaccines, Romanian HPV vaccination programs and cervical cancer screening programs. For instance, the “National Screening Program for Cervical Cancer” is straightly cited in the Materials and Methods section without any former explanation.

Moreover, the section seems harshly truncated lacking a proper description of the aim of the study.

Regarding HPV-related diseases and vaccination topics, I invite the authors to consider the following works:

G. Capra et al. 2017, Potential impact of a nonavalent vaccine on HPV related low-and high-grade cervical intraepithelial lesions: A referral hospital-based study in Sicily. Hum Vaccin Immunother

L. Bosco et al. 2021, Potential impact of a nonavalent anti-HPV vaccine in Italian men with and without clinical manifestations. Sci Rep

G. Capra et al 2022, Human Papillomavirus (HPV) Infection and Its Impact on Male Infertility. Life

Capra, G. et al 2008, HPV genotype prevalence in cytologically abnormal cervical samples from women living in south Italy. Virus research.

RESULTS:

Results must be reported always in the same way, i.e., percentages near to numbers between brackets, or viceversa.

Page 3, lines 102-103: “Also, most cases…”, “For some of those cases…”.

These expressions are too generic, you should report the correct numerical data.

Page 4, lines 109-125: “The cost per CC case treated increased 109 constantly in the last years…”

Page 5, lines 141-146: “As for access to education we need to first say…”

All these sections look more like an interpretation than a description of results; hence it would seem more appropriate to place them in the Discussion.

Page 4, line 135: You should cite Table 1.

Table 2, Page 4: The table structure is quite confusing and hard to read. You should appropriately separate the numbers and align them with the parameters depicted.

Figures 1-12:

Using several pie charts to describe each questionnaire answer is confusing and dispersive. These data could be put in a single table like Table 1. In doing so, I believe that the depiction of the results would appear more direct and easier to read.

DISCUSSION:

I believe that the discussion section is excessively long, with a subdivision into paragraphs that could be avoided. In fact, the whole section could be made more concise by removing or placing in the Introduction some of the information. Some examples of the latter include the description of Romania's situation with respect to other EU countries (Pages 10-11, lines 250-260), EU cervical cancer screening recommendations and European countries screening implementations (Page 11, lines 274-282), as well as the description of HPV and related diseases (Page 14, lines 414-424). You should also leave more space for comments on your results.

Page 14, line 415: “…across at least one strain of the virus…”

You should replace the word “strain” with the correct term “genotypes”.

English language could be improved.

Author Response

Dear Reviewer 2,

We highly appreciate the time you took in order to understand the message we wanted to convey with our study. The suggestions you provided were useful and help us improve our work and we addressed them as follows:

  1. We renamed the article in order to make it easier to read
  2. We reported the correct numerical values for lines 102-103
  3. We corrected lines 109-125 by adding numerical values were possible
  4. We moved lines 141-146 in the Discussion
  5. We cited table 1 in line 135
  6. Table 2 was revised in order to make it easier to read and its layout has been improved.
  7. We removed Figures 1-12 and substitute them by Tables.
  8. We substituted “strain” with genotype in line 415
  9. We reported the results in the same way, and the results were discussed further
  10. The Introduction was extended and now contains additional information on the preventive measures currently available in Romania – screening and vaccination programs.
  11. We added the aim of the paper in the Introduction Section.
  12. Although quite long the discussion portion of our manuscript is necessary in our opinion and we opted for keeping it as such. Further information was added as a result of the other peer-reviews we received.
  13. All references you recommended are extremely interesting and were incorporated in our manuscript. We thank you for your suggestions.
  14. Grammatical and English errors - The document was subjected to multiple grammar and spelling checks and revised by one of our authors which has a Native Speaking Level English Language degree. Furthermore, if deemed necessary we will use MDPI professional checking of our manuscripts before printing.

Reviewer 3 Report

Major revisions:

-        The Authors must add in the introduction section the aim of the study. The addition of data obtained from questionaire is unneccessary. I'd suggest to  the Authors to remove this part of the study.

-        Could the Authors specify which type of prevention strategies (e.g., national screening, voluntary screening, vaccination…) are currently available against cervical cancer in Romania?

-        Line 141-146: this is part of discussion section.

-        Line 266-268: the sentence and percentage are not clear. Should the Authors clarify this point?

-        There is a lack of correspondence between the results obtained and the discussion: epidemiological data are discussed in general but a true discussion, as comparison with other geographical areas, is lacking.

-        The data on the questionnaire, considering the population surveyed and the type of questions, have important limitations. I would suggest that the authors remove this part from the work and keep only the epidemiological part.

-        Much information in discussion section are redundant and disconnected from data collected in hospital databases.

-        An important revision of English language is needed.

Minor:

·        Line 45: Remove the comma after “If”

·        Table 2: is not clear for readers. Please, review the layout or add some graphs that could be clearer for readers.

·        Line 128: change “crevention” in “prevention”.

·        Line 140-141: “(known for lower income and accessibility to education and healthcare” should be remove, it is just specified in the introduction.

The manuscript requires an important revision of English language by a native speaker.

Author Response

Dear Reviewer 3,

We the authors thank you for the time you took in order to do such a rigorous analysis of our manuscript and we are most grateful for your suggestions which will surely allow us to improve our article. After reviewing our material in accordance with your suggestions we also found a few additional small mistakes that we were able to rectify in time, and we are grateful for your contribution.

We have addressed each of your concerns as follows:

  1. Grammatical and English errors - The document was subjected to multiple grammar and spelling checks and revised by one of our authors which has a Native Speaking Level English Language degree. Furthermore, if deemed necessary we will use MDPI professional checking of our manuscripts before printing.
  2. We removed the comma after “If” in line 45.
  3. We revised table 2 in order to make it easier to read and its layout has been improved.
  4. We changed “crevention” in “prevention” in line 128.
  5. We removed “(known for lower income and accessibility to education and healthcare)” in lines 140-141.
  6. We reformulated lines 266-268 in order to clarify this point.
  7. We added the aim of the paper in the Introduction Section. In Introduction we also specified and described the prevention strategies currently available in Romania.
  8. About the inclusion of the questionnaire data in our study, we consider it important in order to give a dimension of the level of knowledge and access to healthcare services in some populations in Romania and should be interpreted as such. Also, in the opinion of the other reviewer, the questionnaire data was considered the interesting part of the article and he/she would have preferred an article containing only that information and not the epidemiologic part, he/she even suggesting that part entirely. Since this is a personal preference of each reviewer, we opted for keeping both parts and trying to accommodate each reviewers’ concerns. We agree that the study population of women participating in screening via the mobile unit is not representative of the Romanian screen-eligible population, and we have acknowledged this as a limitation of our study in Section 5 – Limitations rows 431-442 of unrevised manuscript and suggested that further studies on a more representative population may change the results.
  9. We have rewritten most of the Discussion Section trying to create a true correspondence between the results obtained and the discussion and reduce redundancy. We added a comparison to other geographic areas. We moved lines 141-146 to the Discussion Section of the manuscript as per your suggestion.

Round 2

Reviewer 3 Report

dear Authors,

I would thank you for welcoming the comments and suggested revisions.

I would suggest the Authors a small check of the grammar and extend the comparison to other territorial contexts. In particular, in reference to HPV-related diseases it should be highlighted how the prevalence of HPV infection can vary widely. In this regard, I would add to the statement "HPV chronic infection can lead to head and neck [61-64][58-61], skin [65-68][62-65] or anogenital [69][66] cancers, with different prevalence.." along with the following references: doi: 10.3390/cancers14174205 for Head and Neck cancer, and doi: 10.3390/ijerph17124516 for anal cancer.

 After these minor revisions, the manuscript is adapted to publication.

Minor editing is required

Author Response

Dear Reviewer 3 (round 2),

We highly appreciate the time you took in order to understand the message we wanted to convey with our study. The suggestions you provided were useful and helped us improve our work, and we addressed them as follows:

  1. Grammatical and English errors - The document was subjected to multiple grammar and spelling checks and revised by one of our authors which has a Native Speaking Level English Language degree. Furthermore, if deemed necessary, we will use MDPI professional checking of our manuscripts before printing.
  2. We extended the comparison to other territorial by adding a comparison of Romania to the other EU27 member states. Using data freely available on ECIS (European Cancer Information System) we added Figures 2-4 depicting the cervical cancer situation of Romania vs EU 27 in 2020, the reduction of incidence and mortality in 2020 compared to 2015 and also a projection of cervical cancer in 2040 based on current data for all member states of EU 27.
  1. All references you recommended are extremely interesting and were incorporated in our manuscript. We thank you for your suggestions.
  2. We high-lighted how the HPV related pathologies can vary widely in prevalence and added “with different prevalence...” where you suggested.